# Using Behavioral Characteristics to Design Amphibian Ladders for Concrete-Lined Irrigation Channels

**Bo Bi** [1], **Jian Tong** [2], **Shaohua Lei** [1], **Dan Chen** [3,*], **Qiu Jin** [1], **Dalin Hong** [1], **Xiaojun Wang** [1], **Jing Chen** [3] **and Siyuan Zhao** [3]

1   State Key Laboratory of Hydrology-Water Resources and Hydraulic Engineering, Nanjing Hydraulic Research Institute, Nanjing 210029, China
2   Soil and Water Conservation Monitoring Station of Jiangsu Province, Nanjing 210012, China
3   College of Agricultural Science and Engineering, Hohai University, Nanjing 211100, China
*   Correspondence: cherrydew@hhu.edu.cn; Tel.: +86-025-8582-9689

**Abstract:** Human-dominated landscapes have become a serious threat to amphibian populations worldwide. In such landscapes, concrete structures act as barriers to migration, fragmenting habitat and causing mortality. In China, concrete irrigation channels, which play an important role in agriculture, impede the movement of anuran amphibians. To promote the sustainable development of irrigated agriculture, we performed behavioral experiments to examine the ability of a common Chinese frog species (*Pelophylax nigromaculatus*) of four different body sizes to use corridors along a gradient of six different slopes to escape from irrigation channels. We found that body size was positively related with frogs' ability to climb the ladders. Most frogs could not escape if the slope was ≥65 degrees, but all frogs could successfully navigate a ladder with a slope of 45 degrees. Based on our experimental results, we propose a simple improved design for amphibian ladders that would greatly improve the success of frogs in escaping from irrigation channels. This research is expected to provide scientific reference data and technical support for frog conservation in the study area, as well as the development of ecological restoration of irrigation districts throughout the world.

**Keywords:** frog conservation; aquatic ecosystem; irrigation district; sustainable development; ecological reconstruction

## 1. Introduction

With the rapid development of urbanization construction and agricultural production, the intensity of human exploitation and utilization of soil and water resources has gradually increased. Thus, many native environments have been transformed into human-dominated landscapes, which has caused the loss and fragmentation of wildlife habitat [1–4]. At present, human-induced land-cover changes have damaged ecosystems, which leads to changes in species composition, abundance, and diversity [5,6]. It is widely agreed that competition between socio-economic development and environmental conservation can be intense, and infrastructure projects, including dams, roads, and waterways, have negative ecological effects [7–10]. Most waterways are made of concrete, whose functional needs only focus on the utilization efficiency of land and water resources and slope stability, while the biological conservation is rarely taken into account. Due to the concrete structures, the environment of the organisms living in the fragmented habitat becomes harsh. Many amphibian species are currently undergoing population declines, range reductions, and even face regional extinction [11,12]. Habitat fragmentation is undoubtedly one of the primary causes of amphibian losses [13–15].

Amphibians are a bridge in the process of vertebrate evolution and they assume a vital position in the ecosystem; they have important ecological, scientific and social values [16,17]. For example, frogs have long played an important role in agricultural ecosystems, they are widely distributed in paddy fields, can control pest populations,

increase food production and reduce pesticide use [18–20]. Evidence clearly shows that human factors influence the migration rate of frogs in agricultural and other landscapes, and that substrate composition and weather conditions affect the motility of frogs [17,21,22]. Frogs live on water and land, and therefore, the quality of the interface area is especially critical for their lifecycle. Their behavior is generally restricted by moisture availability and ambient temperature, and they are more sensitive to habitat fragmentation than other terrestrial vertebrates. Under conditions of hot temperature and low humidity, most frogs are inept at jumping and employ a creeping movement; thus, they are easily trapped in the concrete structures [23–25]. However, the behavioral ability of frogs cannot match the structural parameters of concrete-lined irrigation channels.

China is a large agricultural country with >3,000,000 km of irrigation channels [26]. Green and sustainable development of irrigated agriculture and protection of agricultural biodiversity are the inevitable requirements for the construction of modern irrigated districts in China [27,28]. However, in order to ensure the efficient utilization of irrigation water, the construction of irrigation channels has been dominated by hard engineering, which has caused 2–3 amphibians (mostly frogs, but also toads and snakes) to die in every 100 m of irrigation channels in Tongxiang County [29]. Because frogs are active on land and in water, mortality can result when animals fall into concrete waterways, which have low water levels or are empty, and then, they are unable to return to riverbanks [30,31]. It is inferred that more than 10,000 frogs die each year in a large-scale irrigated district (≥200 square kilometers), but the effects of agricultural concrete ditches on frog populations and their seasonal movements are rarely assessed [26]. Frog deaths caused by waterways have only recently aroused extensive concern [27,32].

For many decades, ecological corridors have been employed to safely guide the movement of animals between habitats and reduce the negative impacts of artificial structures [33,34]. Different from the wide range of target species in highway wildlife corridors, the research objectives of waterway wildlife corridors are relatively concentrated, mainly focusing on the barrier effect of dams on migrating fish species (i.e., fishway research). The impact of river engineering construction on amphibian habitats has been neglected for a long time [35]. The construction of amphibian corridors in irrigation channels is an important means to improve the connectivity of the agricultural landscape and the migration efficiency of frogs in farmland. Therefore, researchers and ecological engineers are now working to improve the design of amphibian ladders so that these structures can counter the adverse impacts of concrete-lined irrigation channels on frogs [36,37]. However, these projects are very costly and questions regarding optimal design and whether or not they are effective in conserving frog populations remain unanswered [27].

We focused on testing the behavioral characteristics of captive dark-spotted frogs (*Pelophylax nigromaculatus*, formerly *Rana nigromaculata*) as attempted to escape from concrete-lined irrigation channels in order to find a simple and cost-effective design of local engineered modification. First, this article examines the escape capacity of frogs with four different body sizes when facing the amphibious ladders with six different slope gradients. Second, the article analyzes the frogs' escape behavior and trajectory, and explores the relationship between behavioral capacity and body size to achieve a better understanding of the behavioral characteristics of the frogs. The ultimate goal of our study was to provide necessary information to design a biological corridor to allow anuran amphibians to cross dangerous sections of irrigation channels, thus protecting fauna populations in agricultural landscapes in east-central China.

## 2. Materials and Methods

### 2.1. Study Area

Lianshui County (longitude 119°170847 E, latitude 33°736082 N) is located in the north of Jiangsu Province, China (Figure 1). The county has a temperate monsoon climate, featuring four distinct seasons. The annual precipitation is about 1014 mm, the annual average temperature is approximately 14 °C, and the average annual relative humidity is

77% [38]. The area of arable land in Lianshui County amounted to 10 million ha, including the effective irrigated area of 7 million ha. The main soil type in Lianshui County is sandy loam; thus, concrete-lined irrigation channels are frequently constructed in order to improve agricultural water use efficiency [38].

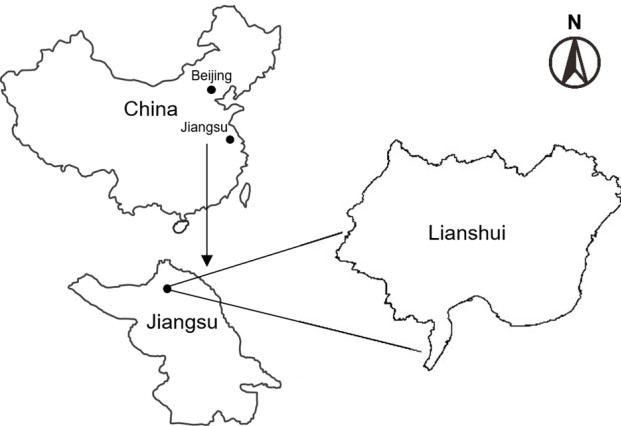

**Figure 1.** Map of the study area.

Lianshui County has a sound irrigation system. In the construction of irrigation engineering, waterways have changed from semi-natural habitats that can connect with farmland to fragmented habitats that cause an ecological barrier between water and land (Figure 2) [38]. The survey results showed that an average of two to three amphibians are found in every 100 m irrigation channels per year due their inability to escape from the concrete channels [29]. It is estimated that at least 10,000 trapped amphibians die every year due to lack of water and exposure to the sun in the concrete-lined irrigation channels (569.52 km) of the study area.

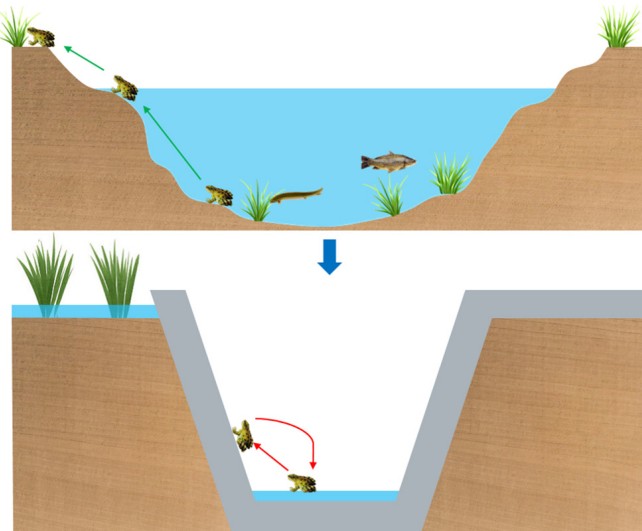

**Figure 2.** Habitat fragmentation of frogs caused by concrete-lined irrigation channels (the red lines indicate that the frog cannot escape from irrigation channels with steep slope, large depth, and smooth surface).

## 2.2. Animal Collection and Rearing

The study focused on the endemic and endangered frog species, *Pelophylax nigromaculatus*, because they are widely distributed in the study area, and can be collected relatively easily. In addition, they are listed as a national protected animal in China, and have important ecological, scientific, and social value [39]. The irrigation cycle and frog life

cycle should be considered comprehensively in this experiment, so biological investigation is needed. *P. nigromaculatus* lives in the still water of paddy fields and ponds. Its adult body length mainly ranges between 7 and 8 cm, and its body weight ranges between 50 and 60 g [40]. There are four different developing periods of the *P. nigromaculatus* life cycle; first, there is an egg, which develops into a tadpole (after 4–8 days); then, a froglet (after 70–78 days), and then, an adult frog (after 90–120 days); the entire growth process takes about 6–7 months to complete [41].

According to the survey and analysis methods of terrestrial wildlife resources, this study adopted the line transect method to collect *P. nigromaculatus* in the field of Lianshui County [42]. The survey was carried out four months after the frogs emerged and entered the breeding period. According to the habitat type, activity range, and ecological habits of frogs in the study area, the line transects were laid in the field. Because the experimenter needed to walk along the transect line for investigation and capture, and their motion range was limited, the single-side width of the transect was 2–5 m. The test line transects were evenly distributed in the study area. Considering the biological rhythms of frogs, sample collections were mainly conducted from 19:00 to 22:00 at night. Forty individuals were caught in August, 2021. The frogs were kept in aquaria that simulated natural habitats. The experiment was completed within 7 days of capture, and the frogs were returned to the original capture sites [14,19,25].

*2.3. Experiment*

This article presented an experimental research on amphibian ladders in end irrigation cannel systems. Because the ecological embankment was applied to the large-scale backbone channels to solve the problem of the habitat fragmentation of frogs in the agricultural landscape that was caused by concrete-lined irrigation channels [26], the riparian vegetation on the ecological slope could connect the land and water, and assist organisms to migrate between these two habitats. However, most field channels built at the end of irrigation systems are inevitably made of concrete. The depths of field concrete channels in Jiangsu Province usually ranged between 40 cm and 90 cm [38]. Thus, we selected a concrete channel with a depth of 90 cm for our research. We recorded the movement of frogs along 6 types of amphibious ladders that varied in their slope gradients: 45, 50, 55, 60, 65, and 70 degrees. Side-slope gradient is an important factor that affects the movement of frogs [19,25,31].

Full-size physical prototypes of amphibious ladders were built for an experiment designed to test the performance of frogs under the 6 slope gradients (Figure 3). They were made of wooden boards and coated with clay to simulate the surface texture of concrete. Ladders were 90 cm high and 30 cm wide, which allowed five frogs to squat side by side. Convex cross-pieces (2 cm × 2 cm) were set vertically along the surface of the corridors at intervals of 10 cm. The frogs could grasp the cross-pieces easily and use them as jump platforms, and increase the surface roughness of amphibious ladders. We were interested in the effects of the body size on the behavioral capacity of frogs. The body weights of the frogs were measured with an electronic scale, and the frogs' snout-vent lengths were measured using a Vernier caliper.

Groups of 5 frogs were put into the bottom of the dry concrete irrigation channels in sequence to test their escape performance when facing the amphibious ladders with varied slope gradients. Only when a frog passed through a ladder with a given lower slope within the specified time (10 min) could the subsequent greater slope be tested. During each trial, the frogs were kept moist with a spray to maintain their best jumping state, and the climbing experiments were conducted at five-minute intervals. The entire process of the trial was videotaped, and the data were obtained by an image processing method. The vertical height of each jump, the number of jumps, and trajectory on the frogs' escape path were recorded, and the mean and standard deviation were calculated.

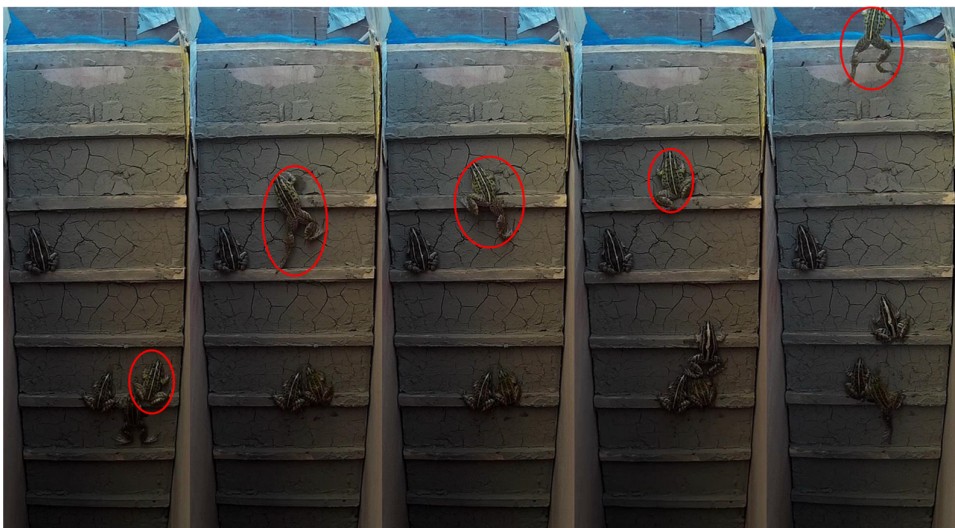

**Figure 3.** The frogs' escape behavior and trajectory in the climbing experiments.

## 2.4. Statistical Analyses

The effects of the body size of frogs and the slope gradient of ladders on frogs' movement capacity were analyzed. Pearson correlation analysis was used to find the relationships between frogs' body length, body weight, and limiting climb gradient. A linear regression model was constructed for the frogs' body length, body weight, and limiting climb gradient. One-way analysis of variance (ANOVA) was used to test the mean difference of the frogs' behavioral ability for 4 different body-size categories and 6 different slope gradients.

## 3. Results

### 3.1. The Limiting Climb Gradient of Frogs and Its Influencing Factors

The limiting climb gradient was defined as the maximum scansorial gradient that an individual frog could climb or jump to escape from the concrete-lined irrigation channels (Figure 4). All frogs could climb corridors with slopes of 45 degrees, whereas 80% of frogs could not successfully navigate a slope of >60 degrees, with only 3 of the 40 frogs negotiating the highest gradient.

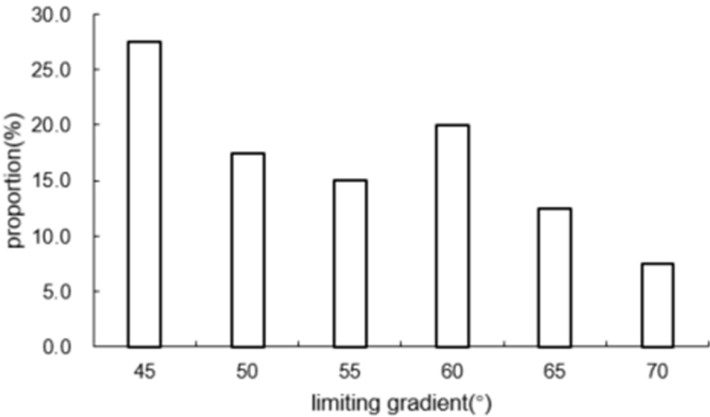

**Figure 4.** The proportion of the limiting climb gradient of all frog samples.

Previous research indicates that frogs with a similar body size have similar abilities to jump or climb, namely the behavioral ability is positively correlated with body size [14,23,25,31]. The body lengths (the snout-vent length) and body weights of the frogs were selected as body-size indicators, and their effects on the limiting climb gradient of the frogs were evaluated. Measurements of body weight (20.9 to 69.9 g, mean = $42.7 \pm 13.7$ g),

body length (7 to 9.2 cm, mean = 7.5 ± 1.0 cm), and limiting climb gradient of all tested frogs are shown in Figure 5. The number of frogs that could successfully escape from concrete-lined irrigation channels via amphibious corridors with slopes of 45, 50, 55, 60, 65, and 70 degrees was 40, 29, 22, 16, 8, and 3, respectively. Additionally, the mean value of the limiting climb gradient of all frog samples is 55 ± 8 degrees.

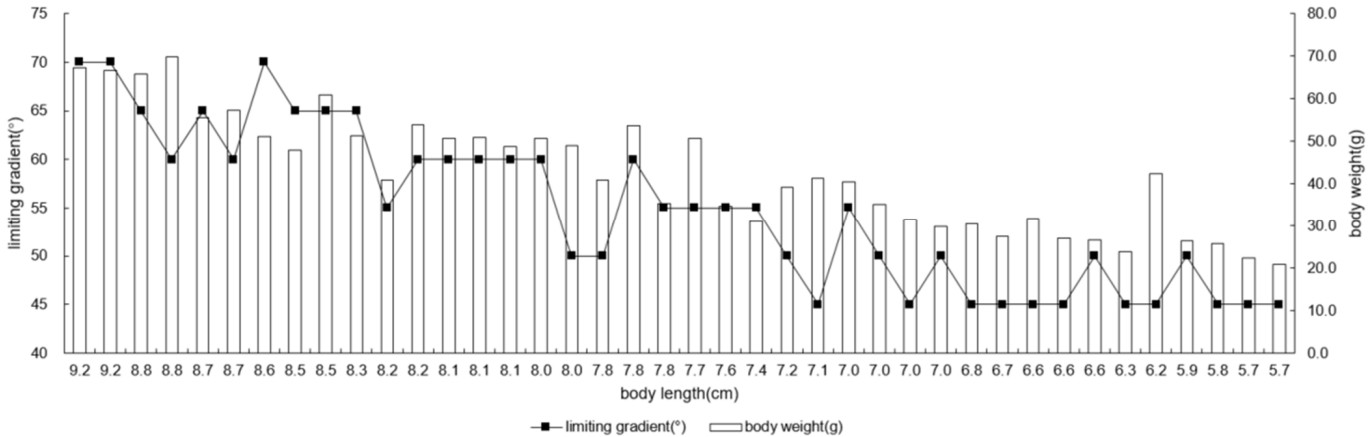

**Figure 5.** Measurements of body weight, body length, and limiting climb gradient of all frog samples.

The Pearson correlation coefficient between body length and body weight was 0.906, and the Pearson correlation between limiting climb gradient and body weight was 0.842, which indicates that there are high positive correlations between the three variables ($r > 0.8$, $p = 0.00 < 0.01$). There is a positive relation between body size and behavioral ability, as larger frogs could navigate steeper climbing gradients and were more likely to escape channels.

Since there were relationships between the three variables, body length (*a*) and body weight (*b*) were selected as the influencing factors of limiting climb gradient (*y*). A predicted relationship between the independent variables and the dependent variable was established, and the linear regression equation was $y = 6a + 0.109b + 4.911$. It is a well-fitting regression model ($R^2 = 0.805 > 0.8$), and body length ($p = 0.00$) has a more significant effect on the maximum scansorial gradient of frogs than body weight ($p = 0.295$). Average body length and weight were substituted into the linear regression model for *a* and *b*, and this equation obtained $y = 55$. This is shown to be consistent with the experimental results of average limiting climb gradient. The adult frogs have body lengths of 7.0–8.0 cm and body weights of 50.0–60.0 g [41]. The body length of 7.0 cm and body weight of 50.0 g were substituted into the linear regression equation to obtain the limiting climb gradient of a relatively small-body-sized adult frog, which is 52 degrees. In addition, the results of the linear correlation analysis indicate that almost all frog samples' limiting climb gradients are higher than 52 degrees. Thus, the great majority of frogs could escape from the concrete-lined irrigation channels via amphibious corridors with a slope of 52 degrees.

### 3.2. Behavioral Characteristic of Frog Escape

Frogs were put into the bottom of the dry concrete irrigation channels with different slope gradients in sequence to test their escape performance. Among the 40 frogs in the sample, 3 of them passed through all six slopes, 5 frogs passed through five slopes, 8 frogs passed through four slopes, 6 frogs passed through three slopes, 7 frogs passed through two slopes, and the remaining 11 frogs only passed through the slope of 45 degrees. The hop-count and the vertical height of each jump were recorded, and the mean and standard deviation were calculated. The average hop-count and the first three jump heights of the frogs that managed to escape are shown in Figure 6.

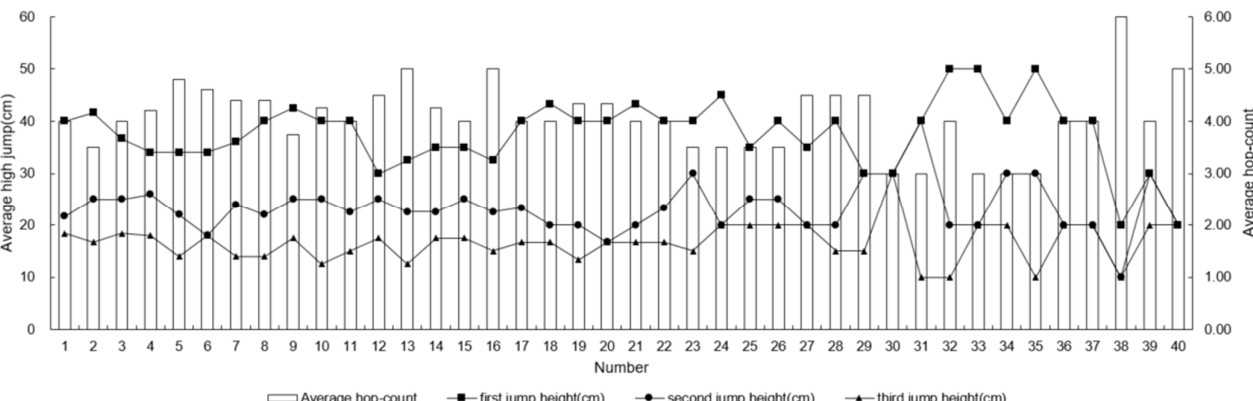

**Figure 6.** Average hop-count and the first three jump heights of the frogs that managed to escape.

The average heights of the frogs' first jump ranged between 20 cm and 60 cm (Figure 7). The frog sample contained 2 frogs whose average first jump heights ranged between 20 cm and 30 cm, for 15 frogs, it ranged between 30 cm and 40 cm, for 20 frogs, it ranged between 40 cm and 50 cm, and for 3 frogs, it ranged between 50 cm and 60 cm. The average heights of the frogs' second jump varied in the range between 10 cm and 50 cm. The frog sample contained 3 frogs whose average second jump heights ranged between 10 cm and 20 cm, for 30 frogs, it ranged between 20 cm and 30 cm, for 6 frogs, it ranged between 30 cm and 40 cm, and for 1 frog, it ranged between 40 cm and 50 cm. The average height of the frogs' third jump varied in the range between 10 cm and 40 cm. The frog sample contained 29 frogs whose average third jump heights ranged between 10 cm and 20 cm, for 10 frogs, it ranged between 20 cm and 30 cm, and for 1 frog, it ranged between 30 cm and 40 cm.

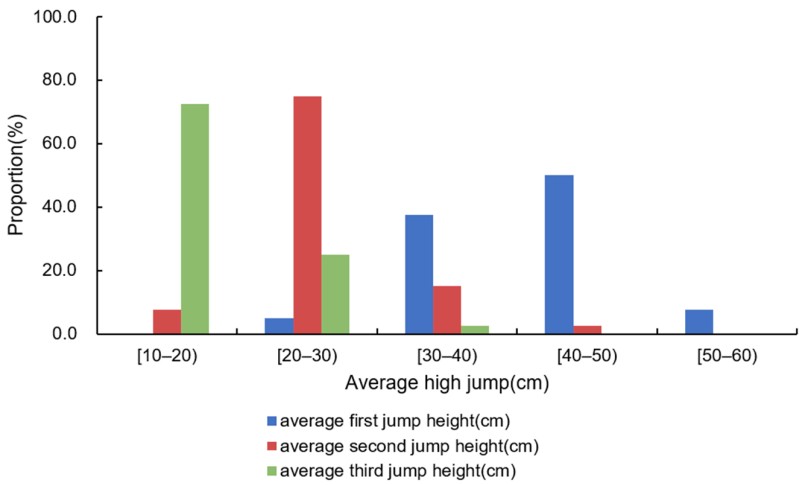

**Figure 7.** The proportion of average high jump of all frog samples.

The average hop-count of the frogs ranged between 3 and 7 (Figure 8). The frog samples contained 11 frogs whose average hop-counts ranged between 3 and 4, for 25 frogs, it ranged between 4 and 5, for 3 frogs, it ranged between 5 and 6, and for 1 frog, it ranged between 6 and 7.

The average hop-count and the first three jump heights of the frogs that managed to escape via ladders with six different slope gradients are shown in Figure 9. The average first jump height of the frogs that managed to escape via the amphibious ladder with a slope of 70 degrees is 46.7 cm, that of the 65-degree slope is 32.5 cm, that of the 60-degree slope is 33.8 cm, that of the 55-degree slope is 37.3 cm, that of the 50-degree slope is 38.6 cm, and that of the 45-degree slope is 38.0 cm. The second jump height averages of the frogs that managed to escape via the amphibious ladders with six different slope gradients were

23.3 cm, 17.5 cm, 21.3 cm, 21.4 cm, 24.8 cm, and 24.0 cm, respectively. The third jump height averages of the frogs that managed to escape via the amphibious ladders with six different slope gradients were 13.3 cm, 16.3 cm, 15.0 cm, 17.3 cm, 16.9 cm, and 16.3 cm, respectively. The hop-count averages of the frogs that managed to escape via the amphibious corridors with six different slope gradients were 3.7, 5.0, 4.8, 4.3, 3.9, and 4.0, respectively.

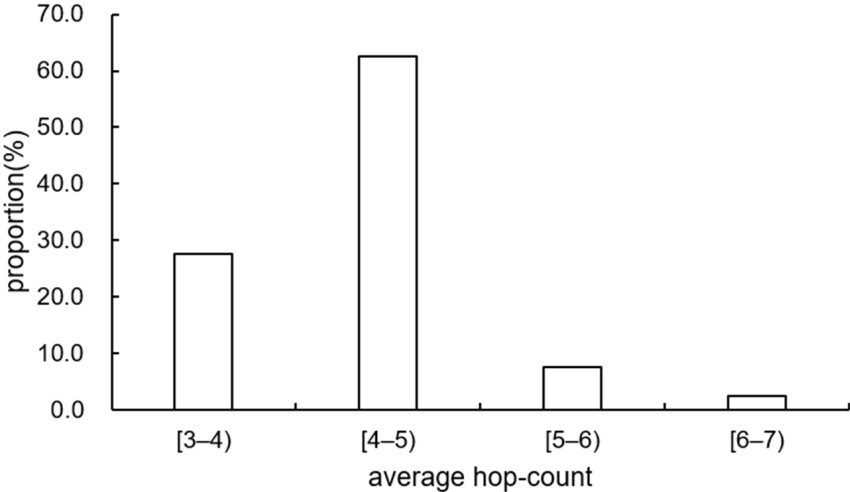

**Figure 8.** The proportion of average hop-count of all frog samples.

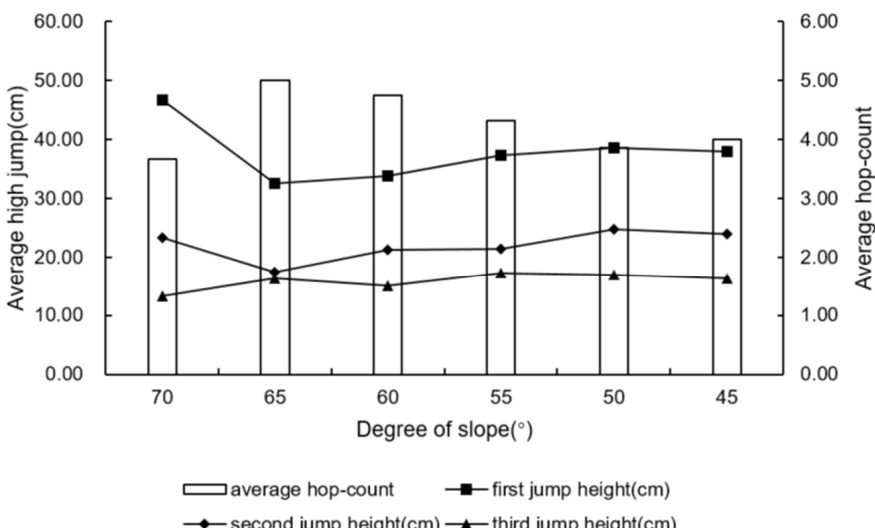

**Figure 9.** Average hop-count and the first three jump heights of the frogs that managed to escape via the amphibious corridors with six different slope gradients.

According to the aforesaid results, body length is the most important factor affecting the behavioral ability of frogs. Thus, the classification standard of body size is body length index, and the frog samples were divided into four groups (A: ≥8.5 cm, B: 84.9–75 cm, C: 74.9–65 cm, and D: <6.5 cm). The frogs in Group A escaped 46 times (n = 9), the frogs in Group B escaped 45 times (n = 13), the frogs in Group C escaped 20 times (n = 12), and the frogs in Group D escaped 7 times (n = 6). The results of the one-way ANOVA of all sample frog's body sizes and escape processes showed that the body size made a big difference in the first jump height of the frogs on the escape process ($p = 0.043 < 0.05$, $F = 2.799$), and made no difference in the hop-count ($p = 0.118$, $F = 2.000$). The larger the body size, the greater the first jump height of the frogs. The results of the one-way ANOVA of the slope gradients of the amphibious corridors and escape processes showed that the slope gradient made a big difference in the hop-count of the frogs on the escape process ($p = 0.011 < 0.05$, $F = 3.149$),

and made no difference in the first jump height ($p = 0.105$, $F = 1.871$). The steeper the ladder slope, the higher the average hop-count of the frogs.

It is shown that body size made a big difference in the first jump height of the frogs, and the slope gradient made a big difference in the hop-count of the frogs. The experimental results showed that the first jump height average of 87.5% of the subjects ranged between 30 cm and 50 cm, the second jump height average of 90.0% of the frogs ranged between 20 cm and 40 cm, the third jump height average of 97.5% of the frogs ranged between 10 cm and 30 cm, and the average of the hop-count of 90.0% of the frogs was less than 5. The first jump height average of the frogs that managed to escape via the corridors with the six different slopes ranged between 32.5 cm and 46.7 cm, the second jump height average of the six different slope gradients ranged between 17.5 cm and 24.8 cm, the third jump height average of the six different slope gradients ranged between 13.3 cm and 17.3 cm, and the average of the hop-counts across the six slope gradients ranged between 3.7 and 5.0. The heights of the successive jumps decreased as the hop-count increased in the escape process of frogs, and the majority of frogs that escaped from the concrete-lined irrigation channels did so in fewer than five jumps.

## 4. Discussion

In this study, we proposed and tested a simple and cost-effective design to allow frogs to escape from agricultural irrigation channels. The full-size physical prototypes of the amphibious ladders with six different slope gradients were built, and the climbing experiments were carried out. The effects of the slope gradients and body sizes on the behavioral patterns of the frogs to escape were examined. Our experiments showed that the lower the gradient of the ladder, the higher the probability that the frogs will escape, in agreement with previous research results [19,25,31]. However, no discussion of the limiting climb gradients for frogs and factors influencing them was provided. It is important to note that a gentler slope will take more farmland, change the structure on channels to a greater extent, and inevitably decrease irrigation efficiency due to reducing the hydraulic smoothness of the channel.

Our results indicate that there is a significant positive correlation in the study species, the dark-spotted frog, between individual frog's body sizes and behavioral abilities. The larger the body size, the greater the ability of frogs to climb steeper gradients, and the greater the likelihood of escape, which is in agreement with the previous research results [14,23,25]. It has been found that there are linear relationships between the frogs' body length, body weight, and limiting climb gradient, and body length has a greater effect on limiting climb gradient than body weight. This finding seems contrary to the conclusions of previous studies that "body weight as the most important influencing factor for the behavioral ability of frogs" [31]. This is probably due to: (1) The species of the frog sample (*P. nigromaculatus*) in this work being different from that of other studies; and (2) The sample size being relatively small.

The observational and experimental data showed indeed that the frogs usually took big jumps by using their hindlimbs in the first half of the escape process, and climbed the convex cross-pieces by using their forelimbs in the second half of the escape process, probably due to physical exhaustion. Therefore, we optimized the design of amphibious corridor according to the frogs' behavioral characteristic in order to make it more targeted to allow frogs to escape. The design method was to set up convex cross-pieces on the surface of the amphibious corridors with slopes less than 52 degrees (Figure 10). It was recommended that the layout distance (vertical height) of convex cross-pieces on the slope surface of the amphibious corridor should be 30 cm, 20 cm, 10 cm, 10 cm, 10 cm, 10 cm from bottom to top, respectively.

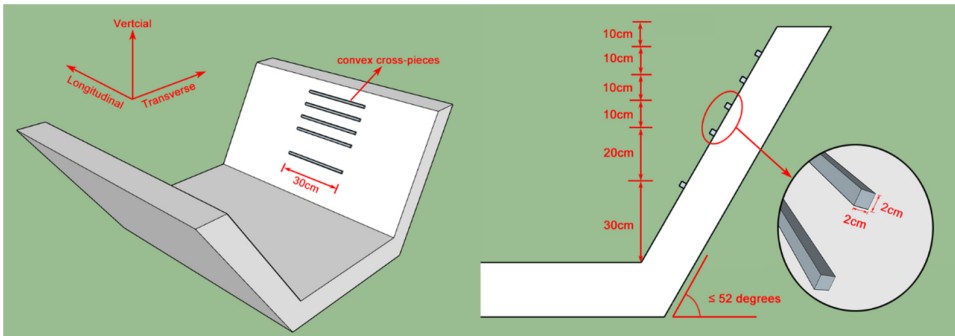

**Figure 10.** Design optimization of the eco-modification for concrete-lining of field irrigation channels.

In future work, we recommend investigating the jumping height, jumping distance, and clinging capability of frogs, analyzing the costs and construction disturbance of engineered modifications, and enlarging the number of individuals and range of species tested. Further studies should also focus on analyzing the effects of temperature and humidity on the behavioral performance of frogs.

## 5. Conclusions

Our study fills a knowledge gap in our capacity to mitigate the effects of concrete-lined irrigation channels on frog migration. Such information is important for the maintenance of biological diversity in agricultural ecosystems in China, as well as for the irrigation district management reform and the implementation of rural revitalization strategies. The designation of amphibian corridors can also help to promote conservation ecology and environmental restoration in China. The key findings of this study are:

1. The habitats and behavioral characteristics of frogs were investigated in order to provide information for designing engineered modifications for the concrete-lining of field irrigation channels. There is a significant positive correlation between the frogs' body size and behavioral ability. A larger body size is better for the frogs to escape. All frogs could escape from slopes of less than 45 degrees, and a slope of 65 degrees did not allow the majority of frogs to escape.
2. There are linear relationships between the two body-size indicators and limiting climb gradient, and body length has a greater effect on the maximum scansorial gradient than body weight. Almost all frogs' limiting climb gradients are higher than 52 degrees, and the number can be used for frog conservation in future slope designs. If construction is limited by the terrain, the recommended gradient of amphibious corridors should be smaller than this number, so as to allow almost all frogs to escape.
3. Given the frogs' behavioral characteristics in the escape experiments, an amphibious corridor design for a common field irrigation channel was proposed. The engineered modification method was to set up convex cross-pieces on the revetment of the concrete-lined irrigation channels, and the recommended layout distance of the convex cross-pieces should be 30 cm, 20 cm, 10 cm, 10 cm, 10 cm, and 10 cm from bottom to top, respectively.

**Author Contributions:** Conceptualization, B.B. and D.C.; methodology, B.B.; software, J.T. and Q.J.; validation, X.W., D.H. and J.C.; formal analysis, B.B.; investigation, D.C.; resources, J.T. and J.C.; data curation, S.Z.; writing—original draft preparation, B.B.; writing—review and editing, D.C. and S.L.; visualization, S.L.; supervision, X.W.; project administration, D.C.; funding acquisition, J.C. and X.W. All authors have read and agreed to the published version of the manuscript.

**Funding:** This research was funded by the National Key Research & Development Program Project (grant number 2017YFC0403205); the National Natural Science Foundation of China (grant number 42101384 & 52121006); the Natural Science Foundation of Jiangsu Province (BK20210043).

**Institutional Review Board Statement:** Not applicable.

**Informed Consent Statement:** Not applicable.

**Data Availability Statement:** Data are contained within the article.

**Acknowledgments:** The present research was conducted at Lianshui Water Conservancy Research Station.

**Conflicts of Interest:** The authors declare no conflict of interest.

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
