# Peer review of "Using Behavioral Characteristics to Design Amphibian Ladders for Concrete-Lined Irrigation Channels"

_sustainability, doi:10.3390/su15076029_

Round 1

Reviewer 1 Report

Using behavioral characteristics to design amphibian ladders for concrete-lined irrigation channels

In the present study, the authors tested how body size and -weight affect the ability of dark-spotted frogs (N = 40) to escape irrigation channels using amphibian ladders of 6 different slopes (ranging from 45 to 70 degrees). The number of jumps and their height was recorded as well as the maximum slope frogs could climb. The authors recommended a ladder with a slope of 52 degree and gave specific recommendations for convex cross-pieces allowing average sized frogs to escape irrigation channels.

I liked the introduction and discussion a lot (clearly written and informative). But I had difficulties understanding the experimental methods and statistics. Data analyses and results show some redundancies, and the results section is unusually long for such a straightforward experiment. I think the manuscript would benefit greatly from condensing the results section, i.e., there are very detailed descriptive statistics in the results section - maybe presenting counts and numbers in a summarizing table instead?

I also have a couple more specific comments, please see below.

Lines 129-130, 146, 148

Grammar.

Lines 135

Was the experiment carried out in the field, using a selected 90° channel or in the lab? I thought trials were carried out in the lab because frogs were captured and kept in aquaria, but this reads like trials were done in the field.

Lines 137-138

So, the effect of slope gradients has been tested before? What does this study do that the others didn’t do (i.e., what does the present study add to the existing literature)?

When calculating the number and height of jumps, did the authors account for the different slope gradients (i.e., were variables controlled for slope gradient)?

Why was the max. trial duration set to 10min? Is this ecologically relevant?

How much time elapsed between successive trials, i.e., frogs were tested repeatedly, how much time did a frog have in between trials?

Lines 151-152

Was the moisturization standardized somehow?

Lines 153

What is the escape path trajectory, i.e., how was it quantified? Please specify.

Mean and standard deviation for each frog and slope gradient?

Lines 159-161

Pearson correlations or linear models? How did those models look like (what is the predictor and response)?

Lines 163-164

What does ‘behavioral ability’ mean?

I don’t think the 4 behavioral categories have been introduced or explained yet.

Figure 4

I don’t understand this Figure. In the main text, the authors say that all frogs could climb the 45 degree ladder but Figure 4 shows that 45 degrees is the limiting gradient for 25% of the frogs.

Lines 179-180

Discussion.

Lines 191-202

Why did the authors do Pearson correlations as well as a linear regression to test for the effects of body size and weight on behavior?

Also, with a size-weight correlation that strong (R > 0.9), I wouldn’t use both variables as predictors in a linear regression because of collinearity. That is, their effects can not be distinguished anymore. It is not surprising that weight didn’t add any explanatory value to the model anymore after size explained a large chunk of the shared variation already. In order to distinguish the two effects, one could do a preparatory linear regression, e.g., with weight as the response and size as the predictor and continue working with the residuals of this regression as values of weight. That is, the residuals represent the variation in weight that is not explained by size, or in other words, it is weight controlled for size.

I think that, strictly speaking, mixed model with frog-group ID as random effect might be appropriate (e.g., a group may contain a particular large or clumsy or aggressive frog that may hinder others from climbing up the latter within the given time limit).

Line 219

Was the average hop count: is it the average number of jumps per individual and slope? Was this standardized for slope gradient?

Lines 224-234

Each frog has one first jump, and one second jump, and so on – right? Or are there several first jumps a frog makes (one on each latter?)? I am confused about a frog having an average first jump. Please specify.

Lines 242-254

That means, Figure 4 is about 3 frogs only, right? Why only analyzing and showing the results of this small sample size that is not representative for the majority of the frogs?

Lines 259-260

Methods.

Lines 262-264 and 265-267

Direction of effect?

Lines 264-265 and 267-268

Strictly speaking, the p-value indicates there is no effect (not a little one).

Lines 298-300

Not so sure about this (see above).

Author Response

Dear Reviewer,

Thank you so much for giving us an opportunity to revise this paper. We are grateful for your detailed comments and suggestions. We believed that your inputs have greatly improved our manuscript. We have considered these comments carefully and made revisions. The manuscript has been revised and re-polished by a native English speaker. Our point-by-point responses to the comments are marked in red.

Reviewer 2 Report

To promote sustainable development of irrigated agriculture, we performed behavioral experiments to examine the ability of a common Chinese frog species (Pelophylax nigromaculata) of a range of body sizes (9.2 to 5.7 cm) to use corridors along a gradient of 6 slopes (70 to 45 degrees) to escape from irrigation channels. The results indicated that body size was positively related with frogs’ ability to climb the ladders, and all frogs could successfully navigate a ladder with slope 45 degrees. There are some comments:

L86: please check the site information.

L88-89: how to get the data?

L103: the legend should be introduced, as what the red line in Fig 2?

The work is important to protact wild life in agricultural ecosystems in China.

Author Response

(The authors gave the same response as above.)

Reviewer 3 Report

I read this manuscript and I found it difficult to understand in many of its parts. Besides the English, the authors did not provide clear and useful information about the topic or their aims. Citations are often not appropriate. The study design was to simply place a ladder with a different inclination to let frogs escape from concrete channels, and find out that the less vertical, the better. Plus, they state that only big-sized frogs were able to escape, and seeing the picture of the ladder there is no wonder because the steps are too far away for small-sized individuals.

Author Response

(The authors gave the same response as above.)

Reviewer 4 Report

Dear Authors,

here is my review for the manuscript entitled "Using behavioral characteristics to design amphibian ladders for concrete-lined irrigation channel" by Bo Bi et al.

Overall, I think that this work can represent a great piece of scientific literature, other than a very useful tool for conservation strategies not only in China, but in every context similar to the one described in this manuscript.

My only suggestion is to provide an extensive revision of the english grammar and lexicon, since there are some mispells, mistakes, and typos in the ms body (e.g., sometimes you type "specie" instead of "species", amd so on).

After such improvement, in my opinion, this ms is suitable for publication.

Best regards.

Round 2

Reviewer 1 Report

I appreciate how very thoroughly the authors replied to all specific comments I made and believe they addressed all points raised. As said previously, I do think the result section would be more compelling if detailed descriptions of sample size and numbers were presented in a table (e.g., 224-233, 243-254) but ultimately, this is of course the authors choice.

My apologies, I’ve made two very ambiguous comments, that led to a misunderstanding; namely „Discussion (lines 179-180).“ and “Methods (lines 259-260).”. I wasn’t asking for a more detailed discussion or for an explanation of the methods, respectively. Instead, I wanted to point out that the text I was referring to represents a discussion and a methods statement and should therefore be presented in those sections instead of being part of the results section. Again, your paper, your choice!

Reviewer 2 Report

this version is better.

Reviewer 3 Report

None